# PolarQuant: Leveraging Polar Transformation for Key Cache Quantization and Decoding Acceleration

**Songhao Wu**[1*]   **Ang Lv**[1*]   **Xiao Feng**[2]
**Yufei Zhang**[2]   **Xun Zhang**[2]   **Guojun Yin**[2†]   **Wei Lin**[2]   **Rui Yan**[1†]
[1] Gaoling School of Artificial Intelligence, Renmin University of China
[2] ShanghaiTech University    [3] Meituan
{songhaowu, anglv, ruiyan}@ruc.edu.cn
fengxiao2023@shanghaitech.edu.cn
{zhangyufei08, zhangxun12, yinguojun02, linwei31}@meituan.com

## Abstract

The increasing demand for long-context generation has made the KV cache in large language models a bottleneck in memory consumption. Quantizing the cache to lower bit widths is an effective way to reduce memory costs; however, previous methods struggle with key cache quantization due to outliers, resulting in suboptimal performance. We propose a novel quantization approach PolarQuant, which provides a new perspective for key cache quantization and efficiently addresses the outlier dilemma. We observe that the distribution of the key states reveals well-structured patterns under polar transformation. Outliers generally appear in only one of the two dimensions, which are rotated together by a specific angle when rotary position embeddings are applied. When represented as two-dimensional vectors, these dimensions exhibit well-organized patterns, with radii and angles smoothly distributed in polar space. This alleviates the channel-wise outliers, making them well-suited for key cache quantization. PolarQuant divides key vectors into groups of two-dimensional sub-vectors, encoding them as the quantized radius and the polar angle, rather than quantizing original key vectors directly. Polar-Quant achieves the superior efficiency in KV cache quantization and accelerates the decoding process by turning the query-key inner product into a table lookup, all while maintaining the downstream performance of full-precision models. Our code is available at https://github.com/ericshwu/PolarQuant.

## 1 Introduction

Large language models (LLMs) have achieved remarkable success across a wide range of real-world applications. As these models evolve, the demand for enhanced long-context capabilities also increases, especially in tasks like contextual retrieval in question answering [17] and reasoning chains generation for complex reflection and decision-making [21, 20]. However, a significant challenge in developing long-context LLMs is the rising memory cost associated with increasing context lengths, which hinders both their practical deployment and further research.

The attention mechanism [4] in LLMs[3]is a major source of computational overhead and memory consumption, which increase rapidly with context length. To reduce this cost, Key-Value cache (KV cache) is a common strategy, which stores and reuses keys and values for generation to avoid the

---

*Equal contribution. Work done during Songhao Wu's internship at Meituan.

†Corresponding authors: Rui Yan (ruiyan@ruc.edu.cn) and Guojun Yin (yinguojun02@meituan.com).

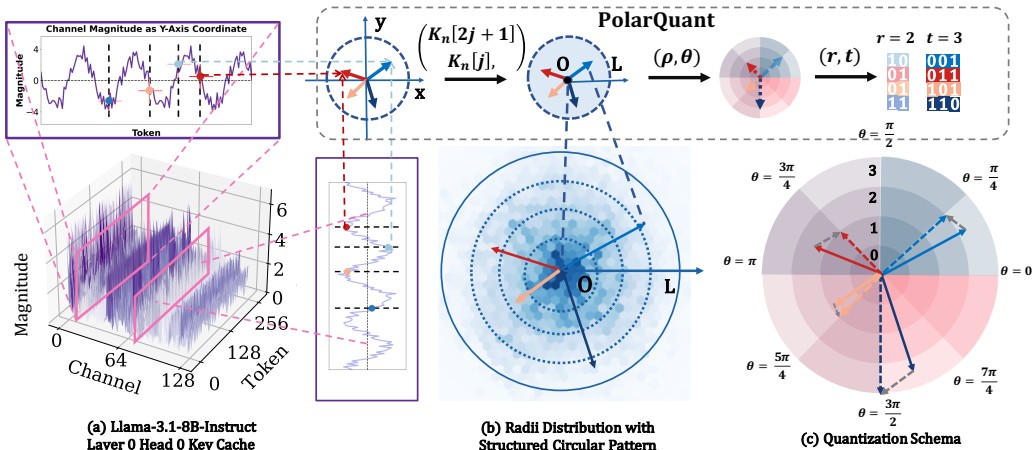

(a) Llama-3.1-8B-Instruct Layer 0 Head 0 Key Cache

(b) Radii Distribution with Structured Circular Pattern

(c) Quantization Schema

Figure 1: (a) Illustration of the activation distribution for the key cache, exemplified by Llama 3.1-8B-Instruct (Layer 0, Head 0). The key cache exhibits channel-wise outliers, where the magnitudes of a few channels significantly larger than others across tokens. We observe that these outliers generally appear in only one of the two dimensions rotated together by RoPE. (b) When viewing these two dimensions in a two-dimensional plane, although the individual x- or y-axis may contain outliers, they collectively form stable circular patterns, making quantization of the original outliers easier. Each blue dot represents a mapped two-dimensional vector, with transparency indicating frequency. (c) PolarQuant using $r = 2$ bits to quantize radii and $t = 3$ bits to quantize polar angles. The colorful arrows indicate sub-vectors formed by pairs of dimensions in the keys; the quantized results are shown with colorful dashed arrows. The quantization error is represented by the grey dashed arrow.

redundant computation. Nevertheless, as the context length increase, the memory required for KV cache can surpass that of the model weights, making it the dominant factor in overall memory usage.

Many solutions have been proposed to reduce the memory cost of KV cache. Some studies introduce memory-efficient attention modules, such as MQA [25], GQA [2] and MLA [9]. While promising, these module architectures require training LLMs from scratch, which limits their applicability. Another research line focuses on the reduction of KV cache size in a compatible manner with pre-trained LLMs. This includes techniques like KV cache eviction [7, 15, 36], which identifies and drops unimportant tokens from the cache , and KV cache quantization [12–14, 19, 37].

This paper focuses on the key cache quantization, which converts the floating-point key cache into low-bit integers to reduce memory usage. In general, key cache quantization is more challenging than value cache due to the presence of channel-wise distributed outliers. Prior studies [13, 19] have highlighted the widespread existence of such outliers in key cache. As shown in Figure 1(a), the key states exhibit larger activations along certain channel dimensions, making token-wise quantization difficult. To address this issue, KIVI [19] proposes a channel-wise quantization strategy that groups and quantizes key elements along the channel dimensions. Building upon this perspective, KVQuant [13] further identifies RoPE as the primary source responsible of the outliers observed in the key cache. The rotation operations in RoPE disturb the magnitude consistency, making accurate quantization complicated. To mitigate this, KVQuant proposes quantizing the keys before applying RoPE, which is described as pre-RoPE quantization. Promising as it is, this approach requires on-the-fly RoPE computation, which consequently introduces potential computational overhead. In this work, we aim to preserve the benefits of pre-RoPE quantization in reducing approximation errors while eliminating redundant computations at each generation step. We propose a polar transformation perspective on handling outliers in the key cache, and effectively address the dilemma in 2D polar coordinates.

KVQuant [13] observes a clear and structured pattern in pre-RoPE key activations: channel-wise magnitude are highly consistent. Recall that RoPE operates a rotation to every two-dimensional

---

[3]In this paper, we focus on decoder-only Transformer-based [29] LLMs using rotary position embedding (RoPE, 26), which are the predominant implementation of advanced LLMs.

sub-vector within the key using an orthogonal $2 \times 2$ rotary matrices.[4] Since rotation is a magnitude-preserving transformation, these 2D sub-vectors inherit the magnitude characteristics seen in the pre-RoPE case. As shown in Figure 1(b), they form well-structured circular patterns when analyzed in 2D polar coordinates. By encoding each sub-vector as its corresponding radius $\rho$ and polar angle $\theta$, the entire key vector can be represented as a collection of all radii and angles. This transformation effectively mitigates outliers, as both the radii and polar angles become smoothly distributed. Building on this, we propose a novel quantization method, PolarQuant, which significantly simplifies the quantization of the key cache. PolarQuant reduces the problem of quantizing key vectors to asymmetrically quantizing $\rho$ and $\theta$ into an $r$-bit and an $t$-bit integer. Intuitively, PolarQuant defines $2^{r+t}$ distinct regions based on $2^r$ angles and $2^t$ radii. Each sub-vector is then encoded by the index of the region it belongs to. Figure 1(c) illustrates PolarQuant for $r = 2$ and $t = 3$.

**PolarQuant achieves superior quantization effectiveness and efficiency over previous methods.** On one hand, polar transformation enables smoother distributions of radii and angles, which alleviates the burden of channel-wise quantization outliers. The superior performance on downstream tasks further demonstrates PolarQuant's superiority in quantization error reduction.

On the other hand, **PolarQuant offers a brand new perspective on key cache quantization, which enables a novel decoding acceleration method.** Unlike pre-RoPE quantization like [13], PolarQuant eliminates the overhead of RoPE recomputation. In the attention mechanism, it replaces the standard query-key multiplication with inner products between two-dimensional query sub-vectors and a quantized polar coordinate representation of key sub-vectors, which have finite and deterministic states. This transforms matrix multiplication to a table lookup, greatly speeding up attention computation. Our contributions are threefold:

(1) We introduce polar transformation for key cache quantization for the first time and derive PolarQuant, a novel and efficient quantization approach;

(2) We propose a new decoding acceleration algorithm as a natural byproduct of PolarQuant. We implement custom Triton kernels to perform fused dequantization and query-key multiplication, which achieves up to $3.18\times$ speedup in long-context generation;

(3) We conduct comprehensive experiments on tasks and models, which further demonstrate the superiority and robustness of our PolarQuant across a wide range of model families and tasks.

## 2 Background

Consider a specific Transformer layer where the input hidden states to the attention block are denoted as $\mathbf{X} \in \mathbb{R}^{L \times D}$, where $L$ is the sequence length and $D$ is the hidden state dimension. For any attention head, the $d$-dimensional query, key, and value vectors are obtained by applying three linear transformations to $\mathbf{X}$. Specifically, for each head $h$, the corresponding computations are as follows:

$$\tilde{\mathbf{Q}} = \mathbf{X}\,\mathbf{W}_Q, \quad \tilde{\mathbf{K}} = \mathbf{X}\,\mathbf{W}_K, \quad \mathbf{V} = \mathbf{X}\,\mathbf{W}_V,$$

where each $\mathbf{W}_* \in \mathbb{R}^{D \times d}$, and the resulting variables have shapes of $\mathbb{R}^{L \times d}$.

The query and key vectors are then applied with RoPE [26] to incorporate positional information. For a query or key vector at position $m \in [1, L]$, the corresponding rotary matrix $\boldsymbol{R}_{m,\Phi} \in \mathbb{R}^{d \times d}$ is defined as:

$$\boldsymbol{R}_{m,\Phi} = \begin{bmatrix} \boldsymbol{r}_{m,\phi_1} & \mathbf{O} & \cdots & \mathbf{O} \\ \mathbf{O} & \boldsymbol{r}_{m,\phi_2} & \cdots & \mathbf{O} \\ \mathbf{O} & \mathbf{O} & \cdots & \boldsymbol{r}_{m,\phi_{d/2}} \end{bmatrix}, \qquad \boldsymbol{r}_{m,\phi_i} = \begin{bmatrix} \cos(m\phi_i) & -\sin(m\phi_i) \\ \sin(m\phi_i) & \cos(m\phi_i) \end{bmatrix}, \quad (1)$$

where $\mathbf{O}$ is a zero matrix, and each $\boldsymbol{r}_{m,\phi_i}$ is a $2 \times 2$ orthogonal matrix. Here, $\phi_i$ is typically defined as $\phi_i = b^{-2i/d}$, where $b$ is the hyperparameter for RoPE base.

This formulation encodes the relative distance $m - n$ between a query at position $m > n$ and a key at position $n$ into their inner product, as shown by:

$$(\tilde{\mathbf{Q}}_m\,\boldsymbol{R}_{m,\Phi})\,(\tilde{\mathbf{K}}_n\,\boldsymbol{R}_{n,\Phi})^\top = \tilde{\mathbf{Q}}_m\,\boldsymbol{R}_{m-n,\Phi}\,\tilde{\mathbf{K}}_n^\top,$$

---

[4]For readers unfamiliar with RoPE, please refer to Section 2.

the keys (after applying RoPE) and values, specifically $\tilde{\mathbf{K}}_n \boldsymbol{R}_{n,\Phi}$ (which we abbreviate as $\mathbf{K}_n$) and $\mathbf{V}_n$, are thus cached to avoid re-computation, known as the KV cache. The KV cache leads to increased memory usage when processing long-context inputs.

Quantization is a simple but effective way to reduce the KV cache size. To clarify the mechanism, we provide a brief overview of key-value quantization below. For the value at position $n$, we follow the token-wise paradigm in [19] and quantize $\mathbf{V}_n \in \mathbb{R}^d$ into $b$-bit, denoted as $Q(\mathbf{V}_n)$.

For an arbitrary dimension $0 \leq j < d$, we have:

$$Q(\mathbf{V}_n[j]) = \texttt{Clamp}\left(\left\lfloor \frac{\mathbf{V}_n[j] - z_n}{s_n} \right\rceil, 0, 2^b - 1\right),$$

where $z_n = \min(\mathbf{V}_n[:])$ is the zero-point, $s_n = (\max(\mathbf{V}_n[:]) - \min(\mathbf{V}_n[:])) \,/\, (2^b - 1)$ is the scaling factor. The colon here denotes iteration over all dimensions, following Python indexing syntax. The function $\texttt{Clamp}(x, min, max)$ restricts $x$ to integers within the range $[min, max]$.

Outliers in key states make per-token quantization challenging, as we discussed earlier and illustrated in Figure 1(a). To address this, previous approaches [19, 13] quantize key vectors channel-wise.

For an arbitrary dimension $j$, a quantized key $Q(\mathbf{K}_n)$ at position $n$ is given by:

$$Q(\mathbf{K}_n[j]) = \texttt{Clamp}\left(\left\lfloor \frac{\mathbf{K}_n[j] - z_{[:]}[j]}{s_{[:]}[j]} \right\rceil, 0, 2^b - 1\right),$$

where the zero-point and scaling factor alternate as:

$$z_{[:]}[j] = \min(\mathbf{K}_{[:]}[j]), \quad s_j = \frac{\max(\mathbf{K}_{[:]}[j]) - \min(\mathbf{K}_{[:]}[j])}{2^b - 1}.$$

Here, the colon in the subscript denotes iteration over all token positions.

## 3 Method

We begin by presenting the key findings of the activation patterns in the key cache (Section 3.1). These insights serve as the foundation for our proposed quantization approach, PolarQuant (Section 3.2).

### 3.1 Motivation

As discussed earlier, outliers in key vectors pose a dilemma for key cache quantization. Our solution to this challenge arises from a key observation:

**Observation:** *When mapping the paired dimensions with outliers to polar coordinates, the resulting 2D vectors show consistent magnitudes. The outliers present in one channel are compensated by the activations of the other, which significantly simplifies quantization.*

Recall that in key vectors, elements in certain dimensions are jointly rotated by the same rotary sub-matrix $\boldsymbol{r}_{n,\phi_i}$. Our analysis shows that the most prominent outliers tend to occur in only one of these dimension pairs.[5]

Figure 1 (b) maps the paired dimensions from Figure 1 (a) onto a 2D Cartesian coordinate system, where the x-axis represents the first dimension and the y-axis represents the second.

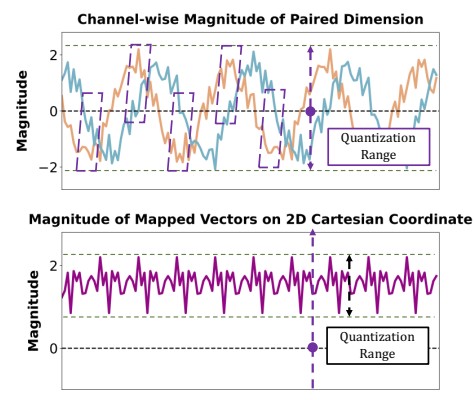

Figure 2: This figure supplements the transformation from Figure 1 (a) to Figure 1 (b), showing how PolarQuant better resolves the outliers.

---

[5]For efficiency, the rotary matrix is typically applied in an element-wise multiplication manner [26]. To simplify implementation, dimensions $i$ and $i + d/2$ are often rotated together, rather than $i$ and $i + 1$. This results in non-adjacent outliers in Figure 1(a), but it does not affect our analysis, which is based on the matrix multiplication formulation (Eq. 1).

Despite large variations in individual x and y values (which would indicate outliers in isolation), the mapped vectors form a well-structured pattern. In other words, when transformed into polar coordinates, the outliers are characterized by a smoothly distributed radial coordinate $r$ and a polar angle $\theta$. This structure significantly alleviates the quantization challenges faced by key caches.

Figure 2 provides a supplementary illustration of how this polar transformation improves quantization. The top figure illustrates a pair of dimensions exhibiting outliers, each with a large individual value range. Quantizing this large range inevitably results in a loss of precision. In the bottom figure, by combining these dimensions into a 2D vector, the norm (i.e., the polar radius) shows a significantly narrowed value range, facilitating quantization.

## 3.2   PolarQuant: Polar-coordinate-based quantization of post-RoPE key cache

Building on these insights, we introduce a novel key cache quantization approach based on polar transformation. Since the benefits of adopting a polar-coordinate have been outlined in the previous subsection, here we focus on the implementation details.

For an arbitrary 2-dimensional subvector $\left(\mathbf{K}_n[2j], \mathbf{K}_n[2j+1]\right)$ in the key cache at position $n$, where $0 \le j < d/2$, we interpret $\left(\mathbf{K}_n[2j], \mathbf{K}_n[2j+1]\right)$ as Cartesian coordinates in the $xy$-plane. This 2D vector is then converted to polar coordinates, where the radius $\rho_n[j]$ is given by:

$$\rho_n[j] = \sqrt{\mathbf{K}_n[2j]^2 + \mathbf{K}_n[2j+1]^2},$$

and the polar angle is:

$$\theta_n[j] = \texttt{atan2}\left(\mathbf{K}_n[2j+1], \mathbf{K}_n[2j]\right) + \pi, \ \ \theta_n[j] \in (0, 2\pi),$$

where $\texttt{atan2}\left(y, x\right)$ returns the angle between the positive x-axis and the point $(x, y)$.

We perform asymmetric group-wise quantization on $\rho_n[j]$ and $\theta_n[j]$, using a group size of $g$, with $r$-bit precision for $\rho_n[j]$ and $t$-bit precision for $\theta_n[j]$:

$$Q(\rho_n[j]) = \texttt{Clamp}\left(\left\lfloor \frac{\rho_n[j] - z_{\rho[:]}[j]}{s_{\rho[:]}[j]} \right\rfloor, 0, 2^r - 1\right),$$

$$Q(\theta_n[j]) = \texttt{Clamp}\left(\left\lfloor \frac{\theta_n[j] - z_{\rho[:]}[j]}{s_{\theta[:]}[j]} \right\rfloor, 0, 2^t - 1\right),$$

where $z_{\rho[:]}[j]$, $z_{\theta[:]}[j]$ are the zero-points and $s_{\rho[:]}[j]$, $s_{\theta[:]}[j]$ are the scaling factors:

$$s_{\rho[:]}[j] = \frac{\max(\rho_{[:]}[j]) - \min(\rho_{[:]}[j])}{2^r}, \quad z_{\rho[:]}[j] = \frac{\max(\rho_{[:]}[j]) - \min(\rho_{[:]}[j])}{2^r},$$

$$s_{\theta[:]}[j] = \frac{\max(\theta_{[:]}[j]) - \min(\theta_{[:]}[j])}{2^t}, \quad z_{\theta[:]}[j] = \frac{\max(\theta_{[:]}[j]) - \min(\theta_{[:]}[j])}{2^t}.$$

Intuitively, PolarQuant partitions the two-dimensional plane into $2^{r+t}$ regions, spanned by $2^r$ radii and $2^t$ polar angles. Each 2D sub-vector of the key vector is then represented by the center of the region in which it resides. Figure 1 provides an illustration of the quantization process. The corresponding Cartesian coordinates in the key vector at dimensions $2j$ and $2j+1$ are then calculated for the quantized representation $\left(Q(\rho_n[j]), Q(\theta_n[j])\right)$, which is formulated as:

$$[\widetilde{\mathbf{K}}_n[2j], \ \widetilde{\mathbf{K}}_n[2j+1]] = \left[\tilde{\rho}_n[j] \cdot \cos\left(\tilde{\theta}_n[j]\right), \ \tilde{\rho}_n[j] \cdot \sin\left(\tilde{\theta}_n[j]\right)\right],$$

where $\tilde{\rho}_n[j]$ and $\tilde{\theta}_n[j]$ are the dequantized radius and polar angle:

$$\tilde{\rho}_n[j] = \left(Q(\rho_n[j]) + \frac{1}{2}\right) \cdot s_{\rho[:]}[j] + z_{\rho[:]}[j], \quad \tilde{\theta}_n[j] = \left(Q(\theta_n[j]) + \frac{1}{2}\right) \cdot s_{\theta[:]}[j] + z_{\theta[:]}[j].$$

### 3.3 Efficient decoding with PolarQuant

In this section, we present PolarQuant's design for query-key multiplication, which incorporates the idea of post-multiplication dequantization to achieve acceleration of the decoding process. We begin by reviewing the conventional approach to dequantized generation. Specifically, during the generation phase, the cached keys must be dequantized before being multiplied by the current query $Q_m$ at position $m$. For each dimension $0 \leq j < d$, we have:

$$\widetilde{\mathbf{K}}_n[j] = Q(\mathbf{K}_n[j]) \cdot s_{[:]}[j] + z_{[:]}[j],$$

where $\widetilde{\mathbf{K}}_n$ denotes the dequantized key, and the inner product is then computed as $\mathbf{Q}_m \cdot \widetilde{\mathbf{K}}_{[:]}$.

The standard dequantization-then-multiplication operation introduces overhead for PolarQuant, as it demands extra computation. We argue that this overhead is redundant. At any dimension $j$, the dequantized outcomes come from a finite set of size $2^{r+t}$. Since the cache size far exceeds this set, it is more efficient to pre-compute and reuse the post-multiplication intermediate results using a lookup table (LUT). This approach of leveraging LUTs has been explored in prior studies [13, 35]. KVQuant adopts an LUT to dequantize the key cache and restores positional information via RoPE recomputation. PolarQuant, in contrast, avoids RoPE recomputation by directly constructing a LUT for QK product on the fly. This is the key insight behind how PolarQuant accelerates decoding.

Specifically, PolarQuant builds its lookup table within each channel by mapping quantized polar coordinates to Cartesian coordinates and computing the dot products with the query sub-vectors. We implement custom Triton kernels to perform fused dequantization and query-key multiplication for efficient GPU execution of PolarQuant. The breakdown time analysis presented in Section 4.2 further demonstrates the effectiveness of our PolarQuant implementation. More implementation details can be found in Appendix A and our released code.

## 4 Experiment

In this section, we evaluate the performance of PolarQuant to highlight its *effectiveness* and *efficiency*. Our empirical results confirm that PolarQuant can be integrated with LLMs, while maintaining near-lossless performance of generative tasks (Section 4.1). We also highlight the speedup achieved by PolarQuant to showcase the superiority of our decoding algorithm and implementation (Section 4.2).

### 4.1 Preserving performance in quantized language and reasoning models

**General setup.** To ensure fair comparisons, we retain the value cache in full precision to avoid potential bias from its quantization in downstream tasks. We evaluate PolarQuant against several quantization baselines built on HuggingFace Transformers codebase [31]. For all group-wise quantization methods, the group size is fixed at $g = 128$. Additional details about the baseline methods and experimental setup are provided in Appendix B. We evaluate all baselines alongside our PolarQuant on a range of models from mainstream model families, including Llama [1, 28] and Qwen [27].

**Quantizing language models.** In real-world applications of language models, the key-value cache often becomes the primary memory bottleneck when processing long-context inputs. We evaluate PolarQuant and baseline methods on LongBench [5], a widely used benchmark for long-context evaluation. Table 1 presents results on Qwen and Llama. There are two advantages of PolarQuant:

1. PolarQuant performs robustly across different model backbones. Quantization is especially challenging for Qwen models, which exhibit extreme channel-wise outliers in their key cache.[6] When applied to these models, token-wise quantization methods — such as per-token Int. and ZipCache — tend to collapse. In contrast, KIVI-4 and PolarQuant constrain the accuracy drop to within 10%. For Llama-3.1-8B-Instruct, PolarQuant even improves average performance under 3-bit quantization, whereas all baseline methods, including KIVI, result in performance degradation.

2. Averaged across all evaluated settings—covering both model families and quantization precisions—PolarQuant achieves the best overall performance preservation.

---

[6]Qwen2.5 language models are configured with attention bias that can introduce outliers in specific channels.

Table 1: Performance comparison of quantization methods on LongBench. Cell colors reflect the degree of performance degradation compared to the backbone model. QJL results for Qwen2.5 are excluded due to incompatibility with its official kernel. Parenthetical values in the last column denote the performance drop of the quantized model relative to its backbone.

| Quantization | Bits | Single Doc. QA | | | Multi Doc. QA | | | Code Completion | | Avg. ↑ |
| | | NtrvQA | Qasper | MF-en | 2Wiki | Hotpot | Musique | Lcc | RepoBench | |
|---|---|---|---|---|---|---|---|---|---|---|
| | | | | | *Qwen-2.5-1.5B-Instruct (128K)* | | | | | |
| *Bf16* | **16** | 19.44 | 37.22 | 49.69 | 32.68 | 41.57 | 23.99 | 41.86 | 48.55 | **38.88** |
| *Int-4* | **4.25** | 5.11 | 8.80 | 10.60 | 11.99 | 5.90 | 32.82 | 24.65 | 22.57 | **15.30 (-23.58)** |
| *ZipCache$_4$* | **4.25** | 4.35 | 8.52 | 12.07 | 14.14 | 17.04 | 7.57 | 23.68 | 21.48 | **13.61 (-25.27)** |
| *KIVI-4* | **4.25** | 19.89 | 36.52 | 49.83 | 32.18 | 41.51 | 22.89 | 40.69 | 48.72 | **36.53 (-2.35)** |
| *PolarQuant$_{44}$* | **4.25** | 19.34 | 36.48 | 50.80 | 32.80 | 43.25 | 23.25 | 38.62 | 46.76 | **36.41 (-2.47)** |
| *Int-3* | **3.25** | 3.37 | 6.96 | 8.31 | 9.53 | 10.79 | 3.64 | 22.32 | 20.69 | **10.70 (-28.18)** |
| *ZipCache$_3$* | **3.25** | 5.28 | 7.81 | 9.38 | 10.30 | 12.66 | 6.36 | 26.53 | 22.72 | **12.63 (-26.25)** |
| *QJL* | **3.13** | | | | | N.A | | | | |
| *KIVI-2* | **3.00** | 18.39 | 36.07 | 47.94 | 32.51 | 42.09 | 23.33 | 39.50 | 45.31 | **35.64 (-3.24)** |
| *PolarQuant$_{33}$* | **3.25** | 19.09 | 35.60 | 49.47 | 32.16 | 43.47 | 23.02 | 35.77 | 44.16 | **35.34 (-3.54)** |
| | | | | | *Llama-2-7B-Chat (4K)* | | | | | |
| *Bf16* | **16** | 18.95 | 21.14 | 37.70 | 30.64 | 27.81 | 6.94 | 58.30 | 52.17 | **31.71** |
| *Int-4* | **4.25** | 18.24 | 21.79 | 37.56 | 29.80 | 26.24 | 8.83 | 57.95 | 52.70 | **31.64 (-0.07)** |
| *ZipCache$_4$* | **4.25** | 19.53 | 19.80 | 36.35 | 31.47 | 26.35 | 8.23 | 58.91 | 51.80 | **31.55 (-0.16)** |
| *KIVI-4* | **4.25** | 18.38 | 21.16 | 37.19 | 31.67 | 26.90 | 7.85 | 58.32 | 51.99 | **31.68 (-0.03)** |
| *PolarQuant$_{44}$* | **4.25** | 18.40 | 21.37 | 35.05 | 30.18 | 27.92 | 8.59 | 58.82 | 51.95 | **31.54 (-0.17)** |
| *Int-3* | **3.25** | 16.51 | 21.41 | 36.59 | 29.34 | 27.59 | 9.74 | 57.75 | 51.42 | **31.29 (-0.42)** |
| *ZipCache$_3$* | **3.25** | 18.73 | 20.11 | 34.32 | 28.50 | 26.53 | 8.39 | 57.51 | 51.42 | **30.69 (-1.02)** |
| *QJL* | **3.13** | 19.13 | 20.51 | 35.93 | 30.74 | 25.60 | 5.79 | 57.79 | 50.92 | **30.80 (-0.91)** |
| *KIVI-2* | **3.00** | 18.79 | 20.46 | 35.51 | 27.52 | 26.36 | 8.12 | 56.82 | 50.26 | **30.48 (-1.23)** |
| *PolarQuant$_{33}$* | **3.25** | 19.75 | 18.26 | 35.47 | 31.15 | 26.60 | 7.68 | 58.26 | 52.41 | **31.20 (-0.51)** |
| | | | | | *Llama-3.1-8B-Instruct (128K)* | | | | | |
| *Bf16* | **16** | 31.38 | 46.65 | 56.81 | 49.46 | 57.85 | 32.63 | 62.88 | 56.43 | **49.26** |
| *Int-4* | **4.25** | 31.76 | 45.49 | 56.47 | 49.52 | 57.67 | 32.82 | 63.17 | 55.46 | **49.05 (-0.21)** |
| *ZipCache$_4$* | **4.25** | 32.26 | 45.97 | 56.77 | 49.50 | 58.67 | 33.03 | 63.41 | 55.97 | **49.45 (+0.19)** |
| *KIVI-4* | **4.25** | 31.23 | 47.15 | 57.14 | 49.14 | 58.05 | 32.67 | 63.05 | 56.45 | **49.36 (+0.10)** |
| *PolarQuant$_{44}$* | **4.25** | 31.36 | 46.78 | 56.72 | 49.47 | 58.54 | 32.23 | 63.28 | 56.73 | **49.39 (+0.13)** |
| *Int-3* | **3.25** | 29.66 | 45.07 | 55.15 | 49.79 | 58.31 | 32.73 | 60.56 | 54.80 | **48.26 (-1.00)** |
| *ZipCache$_3$* | **3.25** | 31.98 | 44.70 | 55.61 | 49.16 | 58.33 | 31.53 | 61.93 | 54.19 | **48.43 (-0.83)** |
| *QJL* | **3.13** | 32.41 | 44.75 | 56.18 | 48.50 | 57.34 | 32.07 | 61.66 | 55.99 | **48.61 (-0.65)** |
| *KIVI-2* | **3.00** | 31.90 | 45.39 | 54.96 | 49.88 | 58.08 | 32.25 | 62.03 | 54.93 | **48.68 (-0.58)** |
| *PolarQuant$_{33}$* | **3.25** | 32.49 | 46.72 | 56.56 | 49.96 | 58.33 | 32.20 | 63.45 | 56.54 | **49.53 (+0.27)** |

Beyond long-context processing, we also apply PolarQuant on LLMs to examine their generative abilities with normal-length inputs. To explore how quantization affects the emergent capabilities of LLMs, such as chain-of-thought reasoning [30] and in-context learning [6], we benchmark PolarQuant's performance on the 5-shot CoT GSM8K [8]. The results, presented in Table 2, show that PolarQuant effectively supports both short- and long-context inputs, without compromising performance on reasoning or knowledge-intensive tasks.

Table 2: Evaluations on 5-shot CoT GSM8K.

| Llama-2 7B-Chat | Quantization | Bf16 | Int-4 | ZipCache-4 | KIVI-4 | PolarQuant$_{44}$ |
|---|---|---|---|---|---|---|
| | **Acc. (Bits)** | 20.92 (16) | 19.79 (4.25) | 23.12 (4.25) | 21.61 (4.25) | 22.61 (4.25) |
| Llama-3.1 8B-Instruct | Quantization | Bf16 | Int-4 | ZipCache-4 | KIVI-4 | PolarQuant$_{44}$ |
| | **Acc. (Bits)** | 78.85 (16) | 76.35 (4.25) | 78.70 (4.25) | 78.32 (4.25) | 78.77 (4.25) |

**Quantizing reasoning models.** Large reasoning models (LRMs) exhibit remarkable capability in solving complex problems by long chains of thought. Recent studies [18] have shown that complex tasks—such as mathematics [16, 8] and reasoning [23]—are sensitive to the accumulation of quantization errors. Given that LRMs rely on both long-context processing and the generation of lengthy outputs, quantization techniques are particularly critical for their application. Moreover, the

underlying mechanisms of LRMs differ from those of LLMs, making quantization more challenging and offering valuable insights into the effectiveness of different approaches.

We apply key quantization to the distillation-based reasoning models of DeepSeek-R1 [20]. The accuracy scores across different quantization methods and tasks are presented in Table 3. The quantized reasoning models of PolarQuant achieve substantial improvements over the baselines, which provides strong supplementary evidence of PolarQuant's superiority.

Table 3: Overall performance comparison of quantized DeepSeek-R1-Distill models across various reasoning benchmarks. Results for *ZipCache* on Qwen2.5 are omitted due to its performance collapse. Cell colors represent the degree of performance degradation compared to the backbone model. Parenthetical values in the last column denote the performance drop of the quantized model relative to its backbone. Best results are highlighted in bold.

| Quantization | AIME | | | MATH | GPQA | Overall ↑ |
| | AIME24 | AIME25 | AVG. | | | |
|---|---|---|---|---|---|---|
| | *DeepSeek-R1-Distill-Qwen-1.5B* | | | | | |
| *Bf16* | **36.67** | **23.33** | **30.00** | **85.20** | **39.90** | **51.70** |
| *ZipCache-4* | | | N.A | | | |
| *KIVI-4* | 20.00 | 23.33 | 21.67 | 80.40 | 33.84 | 45.30 (-6.40) |
| *PolarQuant$_{44}$* | 30.00 | 20.00 | 25.00 | 80.20 | 37.88 | **47.69 (-3.31)** |
| | *DeepSeek-R1-Distill-Llama-8B* | | | | | |
| *Bf16* | **50.00** | **36.67** | **43.33** | **91.20** | **51.52** | **62.01** |
| *ZipCache-4* | 43.33 | 43.33 | 43.33 | 91.60 | 48.48 | 61.13 (-0.88) |
| *KIVI-4* | 43.33 | 33.33 | 38.33 | 89.80 | 51.01 | 59.71 (-2.30) |
| *PolarQuant$_{44}$* | 60.00 | 36.67 | 48.33 | 89.00 | 50.00 | **62.44 (+0.43)** |

## 4.2 Superior efficiency of PolarQuant

To evaluate the efficiency of the customized decoding algorithm, we provide a comprehensive time breakdown analysis of our PolarQuant implementation. In all experiments, we use the Llama-3.1-8B-Instruct model configuration, which includes 32 query heads of dimension 128, and 8 key/value heads (grouped-query attention [2]). We benchmark the latency of our tailored kernel implementation, as well as the end-to-end generation throughput.

**Latency for query-key multiplication kernel.** We make comparisons of the query-key multiplication implementations for LLM decoding. Specifically, we evaluate the wall-clock latency across different sequence lengths and batch size settings. We benchmark the runtime by summing across 10K iterations of the calculations, and report results for: *Fp16*, *KIVI-4*, *KIVI-2*, *PolarQuant$_{44}$* and *PolarQuant$_{33}$*. Figure 3 illustrates the performance comparison among kernel implementations, while Table 4 reports the multiplication latency with a batch size of 8. We see that, PolarQuant achieves up to **2.7×** speedup compared with KIVI and **1.6×** speedup compared with Fp16 Torch implementation.

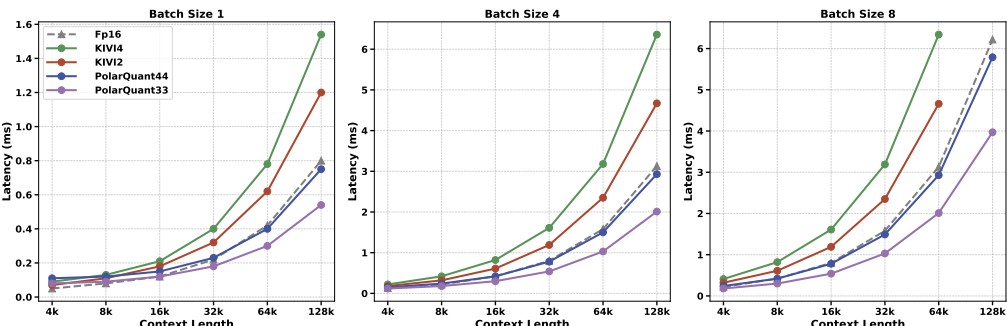

Figure 3: Latency comparisons of PolarQuant across varying batch sizes and context lengths

**Throughput comparison.** We incorporate our custom kernels into the generation pipeline and measure their end-to-end performance. We determine the maximum supported batch size with a sequence length of 32K tokens, with Hugging Face's implementation. We fix the input length at 256 tokens and measure throughput across different generation lengths. Table 4 presents the results; entries marked with † indicate the application of 2-bit value quantization. As is shown in Table 4, PolarQuant achieves up to **3.18×** throughput improvements, and demonstrates significant improvements over KIVI in both latency and throughput. Values in parentheses denote the speedup.

Table 4: Latency, throughput and memory usage comparisons.

| Operation | Latency (s) | | | |
|---|---|---|---|---|
| | **4K** | **8K** | **32K** | **128K** |
| *Fp16* | **0.22** | **0.42** | **1.58** | **6.22** |
| *KIVI-4* | 0.41 (0.54×) | 0.82 (0.51×) | 3.19 (0.50×) | N.A |
| *PolarQuant$_{44}$* | 0.24 (0.92×) | 0.42 (1.00×) | 1.49 (1.06×) | 5.79 (1.07×) |
| *KIVI-2* | 0.32 (0.69×) | 0.61 (0.69×) | 2.35 (0.67×) | N.A |
| *PolarQuant$_{33}$* | 0.18 (1.22×) | 0.30 (1.40×) | 1.03 (1.53×) | 3.97 (1.57×) |

| Configuration | TP. (token/s) / Mem. (GB) | | | |
|---|---|---|---|---|
| | **4K** | **8K** | **16K** | **32K** |
| *Fp16* | **119.1 / 20.78** | **84.5 / 25.21** | **53.6 / 34.07** | **15.3 / 51.79** |
| *KIVI-4* | 129.0 (1.09×) / 19.03 | 96.9 (1.15×) / 21.88 | 65.1 (1.21×) / 27.59 | 39.0 (2.54×) / 38.96 |
| *PolarQuant$_{44}$* | 138.8 (1.17×) / 19.04 | 108.5 (1.28×) / 21.90 | 75.9 (1.42×) / 27.61 | 46.8 (3.05×) / 39.06 |
| *KIVI-2* | 133.2 (1.12×) / 18.89 | 99.9 (1.18×) / 21.55 | 67.7 (1.26×) / 26.91 | 40.5 (2.64×) / 37.62 |
| *PolarQuant$_{33}$* | 144.8 (1.22×) / 19.04 | 111.1 (1.31×) / 21.90 | 78.6 (1.46×) / 27.61 | 48.7 (3.18×) / 39.06 |
| *KIVI-4$^{†}$* | 129.0 (1.09×) / 17.08 | 115.8 (1.37×) / 17.99 | 90.8 (1.69×) / 19.92 | 30.13 (1.97×) / 23.60 |
| *PolarQuant$_{44}^{‡}$* | 144.1 (1.21×) / 17.06 | 128.0 (1.51×) / 17.97 | 111.6 (2.08×) / 19.81 | 46.86 (3.06×) / 23.46 |

## 5 Discussions

### 5.1 Ablation studies of PolarQuant

**Effect of group size $g$.** PolarQuant applies group-wise quantization along the channel dimension. We conduct ablation studies to investigate the impact of the group size $g$ on model performance.

Table 5: Ablation study of group size $g$ on LongBench.

| | **Group Size** | 32 | 64 | 128 | 256 |
|---|---|---|---|---|---|
| **LongBench** | **KIVI-4** | 49.48 (5.00) | 49.47 (4.50) | 49.36 (4.25) | 49.52 (4.13) |
| (Bits) | **PolarQuant$_{44}$** | 49.50 (5.00) | 49.33 (4.50) | 49.39 (4.25) | 49.58 (4.13) |

Specifically, we benchmark PolarQuant and KIVI on LongBench using the Llama-3.1-8B-Instruct model. The results, presented in Table 5, indicate that PolarQuant performs competitively with KIVI across all tested group sizes. It is noteworthy that the quantization parameters occupy $32/g$ bits; therefore, smaller values of $g$ result in higher average bit widths. To strike an optimal balance between performance and parameter overhead, we set $g$ as 128 throughout this paper.

**Impact of RoPE configuration.** We conduct experiments to test the sensitivity of PolarQuant to different RoPE configurations and draw the following conclusions:

(1) PolarQuant exhibits consistent performance across LLMs with different RoPE base frequencies (see Table 1). The experimental results for basic values of $\{10000, 500000, 1000000\}$, highlight PolarQuant's insensitivity to the choice of base frequency.

(2) PolarQuant is adaptable to different RoPE variants. We employ NTK RoPE scaling [24] to extend the LLM's context and apply PolarQuant to the key cache. Critically, mo significant performance drop is observed. We provide detailed experimental results and setups in Appendix C.

**Bitwidth allocation for radii and polar angles.** In PolarQuant, we assign bitwidth asymmetrically between radii and angles. To determine which component requires higher precision, we conduct ablation studies on various bitwidth configurations, enabling a more flexible allocation strategy.

Table 6: Ablation study on asymmetrical bitwidth allocation in PolarQuant.

| **LongBench** | **Bits** | $(\mathbf{r5, t3})$ | $(\mathbf{r4, t4})$ | $(\mathbf{r3, t5})$ | **Bits** | $(\mathbf{r4, t2})$ | $(\mathbf{r3, t3})$ | $(\mathbf{r2, t4})$ |
|---|---|---|---|---|---|---|---|---|
| | 4.25 | 49.41 | 49.39 | 49.51 | 3.25 | 47.66 | 49.53 | 49.00 |

We take Llama-3.1-8B-Instruct as backbone and benchmark PolarQuant on LongBench, to assess the impact of different bitwidth settings. From Table 6, we have the following observations:

***Observation* 1:** Angle quantization is more sensitive to bitwidth. Allocating fewer than 3 bits to angles often results in a significant drop in performance.

***Observation* 2:** Despite the asymmetry in bitwidth allocation, a more balanced distribution between radii and angles can still achieve strong performance.

## 5.2 Compatibility with existing KV cache compression techniques

In this section, we explore the integration of existing KV compression techniques with PolarQuant, which allows for further reductions in KV cache memory occupation. We present the experimental results of Llama-3.1-8B-Instruct on LongBench. As stated in Section 4.1, we retain the value cache in full precision, as key cache is more sensitive to low-precision quantization. Appendix D provides further analysis to support this. We combine value quantization with PolarQuant to verify its compatibility. A standard token-wise quantization is applied to the value cache, consistent with KIVI [19]. Table 7 presents the results. The introduction of value quantization results in only marginal performance degradation, even at 2-bit precision.

Table 7: LongBench results of PolarQuant with value cache quantization .

| Quantization | Value Bits. | NtrvQA | Qasper | MF-en | 2Wiki | Hotpot | Musique | Lcc | RepoBench | Avg. |
|---|---|---|---|---|---|---|---|---|---|---|
| | **16** | 31.36 | 46.78 | 56.72 | 49.47 | 58.54 | 32.23 | 63.28 | 56.73 | **49.39** |
| PolarQuant$_{44}$ | **4** | 31.67 | 46.51 | 56.48 | 49.74 | 58.48 | 32.41 | 63.50 | 56.59 | **49.42 (+0.03)** |
| | **2** | 31.26 | 46.69 | 57.59 | 47.88 | 58.51 | 32.98 | 63.24 | 55.95 | **49.26 (-0.13)** |

We further explore the integration of PolarQuant with token-eviction strategies [15]. As shown in Table 8, PolarQuant does not exhibit significant performance degradation. We left it for future work to combine PolarQuant with existing mixed-precision quantization for further memory saving [3, 10, 33].

Table 8: LongBench Evaluations of PolarQuant with SnapKV.

| LLM | NtrvQA | Qasper | MF-en | 2Wiki | Hotpot | Musique | Lcc | RepoBench | Avg. |
|---|---|---|---|---|---|---|---|---|---|
| *Full KV* | 31.36 | 46.78 | 56.72 | 49.47 | 58.54 | 32.23 | 63.28 | 56.73 | **49.39** |
| *SnapKV*: **4096** | 31.31 | 46.51 | 57.03 | 49.71 | 57.99 | 32.79 | 62.90 | 55.75 | 49.25 (-0.14) |
| *w. PolarQuant* | 31.21 | 46.38 | 56.45 | 49.38 | 58.05 | 31.99 | 62.90 | 55.75 | 49.01 (-0.38) |
| *SnapKV*: **1024** | 31.51 | 43.45 | 56.14 | 49.61 | 57.78 | 32.01 | 62.45 | 56.00 | 48.62 (-0.77) |
| *w. PolarQuant* | 31.27 | 42.61 | 55.18 | 48.60 | 57.49 | 31.12 | 61.54 | 54.57 | 47.80 (-1.59) |

## 6 Conclusion

In this paper, we view the outliers in the key cache of LLMs from a novel polar-coordinate-based perspective, which provides an efficient and effective solution, PolarQuant, to reduce the complexity and quantization costs in previous methods. PolarQuant well preserves downstream performance even in long-context understanding and long chain-of-thought generation, comparable to previous works under 4-bit precision while achieving superior efficiency. We hope the polar coordinate view can inspire the community to advance new low-bit precision quantization techniques.

## Acknowledgments

This work is supported by Meituan through Agentic system X Program. Songhao Wu is supported by "Qiushi Academic-Dongliang" Project of Renmin University of China (No. RUC24QSDL015). We appreciate all reviewers for their valuable comments and suggestions, which are crucial for improving our work. We are also grateful to Xintong Qiu, Jixiang Hong, Tao Tan, Jia-Nan Li and Yuxuan Liu for their insightful suggestions on the manuscript.

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

# A Accelerated Query-Key Inner Product with Cached Keys in PolarQuant

For a quantized representation $\left(Q(\rho_n[j]), Q(\theta_n[j])\right)$, we first dequantize the polar coordinates back into Cartesian coordinates, which serves as the at dimensions $2j$ and $2j+1$:

$$\begin{bmatrix} \widetilde{\mathbf{K}}_n[2j] \\ \widetilde{\mathbf{K}}_n[2j+1] \end{bmatrix}^\top = \begin{bmatrix} \tilde{\rho}_n[j] \cdot \cos\left(\tilde{\theta}_n[j]\right) \\ \tilde{\rho}_n[j] \cdot \sin\left(\tilde{\theta}_n[j]\right) \end{bmatrix}^\top,$$

where $\tilde{\rho}_n[j]$, $\tilde{\theta}_n[j]$ is the dequantization for $(\rho, \theta)$ respectively. $\tilde{\rho}_n[j]$ and $\tilde{\theta}_n[j])$ are formulated as:

$$\tilde{\rho}_n[j] = \left(Q(\rho_n[j]) + \frac{1}{2}\right) \cdot s_{\rho[:]}[j] + z_{\rho[:]}[j], \quad \tilde{\theta}_n[j] = \left(Q(\theta_n[j]) + \frac{1}{2}\right) \cdot s_{\theta[:]}[j] + z_{\theta[:]}[j].$$

$\left(Q(\rho_n[j]), Q(\theta_n[j])\right)$ represents $\left[\widetilde{\mathbf{K}}_n[2j], \widetilde{\mathbf{K}}_n[2j+1]\right]$ as a state in the lookup table.

When computing the query-key inner product for the dimensions $2j$ and $2j+1$, the result is:

$$\mathbf{product}_{2j,2j+1} = \mathbf{Q}_n[2j] \cdot \widetilde{\mathbf{K}}_n[2j] + \mathbf{Q}_n[2j+1] \cdot \widetilde{\mathbf{K}}_t[2j+1],$$

and the final inner product is the sum:

$$\sum_{0 \leq j < \frac{d}{2}} \mathbf{product}_{2j,2j+1}.$$

Figure 4 presents a naive PyTorch [22] implementation of the aforementioned dequantization-and-multiplication operation. We implement a fused Triton kernel that reproduces the functionality of the PyTorch code for improved computational efficiency. More details can be found in our code.

```python
import torch

def attention_decode_forward_pytorch_impl(
    q,  # q's shape: (B, N, 1, 2, D)
    r, rscale, rmn,  # r's shape: (B, N, G, D)
    t, tscale, tmn,  # t's shape: (B, N, G, D)
    # rscale, rmn, tscale and tmn's shape: (B, N, 1, 1, D)
    rbits: int = 4, tbits: int = 4,
):
    # phi: finite set for theta
    phi = torch.arange(0, 2 ** tbits)[None, None, :, None, None]
    phi = (2 * phi + 1) / 2 * tscale + tmn

    # rho: finite set for rho
    rho = torch.arange(0, 2 ** rbits)[None, None, :, None, None]
    rho = (2 * rho + 1) / 2 * rscale + rmn

    phi = torch.cat([phi.cos(), phi.sin()], dim=-2)

    accumulator = torch.sum(q * phi, dim=-2)  # (B, N, 2^tbits, D)
    accumulator = torch.gather(accumulator, 2,
        t.unsqueeze(-1).expand_as(accumulator))
    accumulator *= torch.gather(rho.squeeze(-2), 2,
        r.unsqueeze(-1).expand_as(accumulator))

    accumulator = accumulator.sum(-1)

    return accumulator
```

Figure 4: Pytorch implementation of the accelerated Query-Key Inner Product in PolarQuant.

# B    Details experiment setup

In this section, we provide additional details about the experimental setups.

## B.1    Setup of baseline methods

This section provides an overview of the baseline methods. Following this, we outline the quantization configurations and variants for both the baselines and our proposed PolarQuant. We also explain how the average number of bits for the quantization parameters is calculated.

**Int-N** applies token-wise N-bit quantization to the key states. This token-wise quantization incurs $32/d$ bits quantization parameters (16 bits for zero-points and 16 bits for scaling factors per token), which amounts to 0.25 bits per token when $d = 128$.

**ZipCache**-N [12] introduces a channel-separable, token-wise scheme for key quantization, where $N$ denotes the quantization bits. Each key channel is normalized by the square root of its maximum magnitude before quantization. Similar to **Int**-N, this method performs token-wise quantization, allocating 0.25 bits for zero-points and scales.

**KIVI**-N [19] employs an asymmetric strategy for KV cache quantization, applying channel-wise quantization to the key cache and token-wise quantization to the value cache. For 4-bit quantization, we use **KIVI**-4 with a group size of 128. For 3-bit quantization, we use **KIVI**-2 with a group size of 32, as the official implementation does not support 3-bit quantization. This channel-wise quantization introduces $(16 + 16)d$ bits of quantization parameters per group, which increases the average bitwidth by $32/g$: 1 bit for $g = 32$ and 0.25 bits for $g = 128$ respectively.

**QJL** [34] applies Johnson-Lindenstrauss transformation to key states, removing the memory overheads associated with storing quantization constants. For a 3-bit quantization schema, this method achieves a key cache bitwidth of 3.13 bits.

**PolarQuant**$_{rt}$ assigns r bits for radii quantization and t bits for polar angles, resulting in $(r + t)/2$ bitwidth for key states quantization. PolarQuant also employs channel-wise quantization with grouping; this group partitioning adds an overhead of $32/g$ bits.

Nearly all methods discussed here require a buffer size or residual length for applying quantization. We exclude the contribution of these residual key states to bit counts for comparisons.

## B.2    Configurations of the open-sourced LLMs

• **Qwen-2.5-1.5B-Instruct** [27] is the instruction-tuned 1.5B Qwen2.5 model, which supports a context length of up to 131,072 tokens and features a base RoPE frequency of 1,000,000.

• **Llama-2-7B-Chat** [28] has a context length of 4096 and base RoPE frequency of 10,000.

• **Llama-3.1-8B-Instruct** [1] is an 8 billion parameter language model, designed to handle a context length of up to 131,072 tokens, and has the base RoPE frequency set to 500,000.

## B.3    Reasoning model evaluation

For fair comparison, the evaluation code is built on the Huggingface Lighteval framework [11]. We use the default generation configuration for datasets splitting, and the EM score is reported in Table 3. More implementation details can be found in our released code.

# C    NTK RoPE scaling experiment

We adopt NTK RoPE scaling [24] to extend the context window of the Llama-2-7B-Chat model from 4096 to 8192. Specifically, we implement dynamic RoPE updates based on the Hugging Face codebase. PolarQuantachieve an average performance of 32.44 on LongBench. Compared to the 32.15 score of the Bf16 baseline, PolarQuant's performance remains competitive.

# D  Sensitivity Analysis of Key-Value Quantization

In Section 5.2, we combine 4-bit key quantization PolarQuant$_{44}$ with 2-bit value quantization and observe minimal performance degradation. We further evaluate value quantization (KIVI, group size 128) on LongBench. By retaining the key cache at full precision, KIVI result in no performance drop. However, when the key is quantized to the same bitwidth while the value is kept at full precision, the performance drops significantly. Table 9 presents the results, the notation $(\mathbf{K_b}, \mathbf{V_c})$ denotes key quantization to $b$-bit and value quantization to $c$-bit.

Table 9: Impact of key and value quantization bitwidths on LongBench performance.

| **LongBench** | $(\mathbf{K16}, \mathbf{V16})$ | $(\mathbf{K16}, \mathbf{V4})$ | $(\mathbf{K16}, \mathbf{V2})$ | $(\mathbf{K2}, \mathbf{V16})$ |
|---|---|---|---|---|
| | 49.26 | 49.54 | 49.30 | 47.73 |

# E  Limitation

Although PolarQuant achieves promising results in reducing storage and computational resources, we also discuss the limitations of our current work. This work focuses exclusively on decoder-only Transformer-based large language models (LLMs) that utilize rotary position embedding (RoPE) as the underlying position encoding mechanism. RoPE has become a prevalent choice in many state-of-the-art open-source LLMs. However, the effectiveness of PolarQuant when applied to models with alternative position encoding methods or attention mechanisms remains an open question and warrants further investigation. Furthermore, exploring more recent LLM backbones [32] is necessary, but due to time and computational limitations, we leave this part of the work for future research.

