# OpenReview forum: "PolarQuant: Leveraging Polar Transformation for Key Cache Quantization and Decoding Acceleration"
_NeurIPS.cc/2025/Conference — NeurIPS 2025 poster_

### Official Review · Reviewer_FV6N · 2025-06-22

**Clarity:** 3
**Significance:** 3
**Originality:** 3
**Rating:** 5
**Confidence:** 4

**Summary:**

The paper presents PolarQuant, an approach that applies a polar transform on the KV caches and quantizes the angle and radius.

**Questions:**

1. Can you please explain how the idea can be mixed with rotation-based quantization techniques like [1]? For example, can't we apply Hadamards of size 2 and fuse it into the weights of the previous layers (for W_k and W_v) and then quantize the KV?

2. I found the figures a bit complex and not easy to understand. Can you please polish them and make them easier?

3. Can you provide an ablation on the group sizes during the quantization?

4. I would like to see a breakdown over the runtimes to see the overhead of getting polar coordinates (like atan2 operation) especially during the generation phase which is the main target of KV-cache quantization.

5. Can you please provide more details about the GPU architecture you have yoused for the benchmarking experiments in the paper?

6. Do you have any explanation about the "Observation 1" in line 252? Why the angles need more precision here? Do you have any intuition?

[1] https://arxiv.org/abs/2404.00456

**Ethical Concerns:**

["NO or VERY MINOR ethics concerns only"]

**Final Justification:**

I found the idea of quantizing the angles (instead of the whole vector) for KV-cache quantization quite interesting. This is something that we may be able to use in other quantization schemes as the angles between embedding vectors is the most important piece of information that the model has (and adds during the inference). However, I think my initial score is fair and didn't change it after discussion with the authors.

**Limitations:**

Yes.

**Paper Formatting Concerns:**

I didn't find any issue with the formatting in this paper.

**Quality:**

4

**Strengths And Weaknesses:**

Strengths

- KV-cache compression is an important area for LLM efficiency (especially for long-context inference) and the intuition of using a polar transformation is very clever. I found the idea (and observations) interesting
- The authors provide a real latency evaluation and compare them against the SoTA KV-cache quantization schemes
- Paper is well-written

Weaknesses

- Figures are a bit complex and hard to understand

---

> ### Author Rebuttal · Authors · 2025-07-31
>
> We sincerely thank you for your constructive suggestions and valuable comments! We hope our rebuttal helps address your concerns.
>
> ---
> **W1 & Q2**: Figures need refinement in presentation.
>
> Thank you for your helpful suggestion.
> We will refine Figure 1 and Figure 2 to more effectively showcase our insights and pipeline.
>
> ---
>
> **Q1**: Can PolarQuant combine with the rotation-based weight quantization?
>
> We find your question interesting, but in this work we primarily lies in activation quantization (e.g., KV cache quantization) rather than weight quantization. There are differences between the two domains, both in their core focuses and in the inherent characteristics of weights versus activations. This means we cannot immediately speak to the feasibility of combining PolarQuant with rotation-based weight quantization.
>
> We acknowledge the value of exploring such intersections and need some time to delve into the paper you referenced to gain deeper insights, which may inform and guide the development of our future work. Thank you again for your interest and for bringing this perspective to our attention.
>
> ---
>
> **Q3**: Ablation on the group size.
>
> Thank you for the insightful suggestion! We conduct an ablation study with different group sizes using Llama-3.1-8B-Instruct. The average results across datasets from LongBench are presented below:
>
> | method   | group size |  LongBench (Avg.) |
> |-|-|-|
> | Polar_44 |     32     |      49.50        |
> | Polar_44 |     64     |      49.33        |
> | Polar_44 | 		128     |      49.39        |
> | Polar_44 |    256     |      49.58        |
> | Polar_33 |     32     |      49.68        |
> | Polar_33 |     64     |      49.38        |
> | Polar_33 | 		128     |      49.39        |
> | Polar_33 |    256     |      49.09        |
>
> We find that PolarQuant maintains consistent performance across different group size settings.
>
> Thank you for the suggestion, we will include this discussion in our revised version.
>
> ---
>
> **Q4**: Breakdown time analysis of key cache quantization in PolarQuant.
>
> Thank you for your thoughtful question. Your input is valuable in refining our paper’s presentation.
>
> We perform a time breakdown analysis of PolarQuant during the generation phase (with a batch size of 1, input length of 256, and generation length of 4096):
> | Operation |         |   Time (s) |
> |-|-|-|
> | Decode    |  All    |   91.07    |
> | Quant     |  Others |    0.45    |
> |           |  Tan2   |    0.10    |
>
> From this analysis, we can see that the overhead introduced by the new key quantization is only 0.6% in comparison to the overall generation cost.
>
> We appreciate your suggestion and hope this can clarify your concerns.
>
> ---
>
> **Q5**: Details about the GPU architecture.
>
> We benchmark our experiments on NVIDIA A800-SXM4-80GB. Thank you for the reminder, I will detail the hardware we use in the revised paper.
>
> ---
>
> **Q6**: Why the angles need more precision?
>
> We notice a significant decline in downstream performance when fewer than 3 bits are assigned to the angles.
> As indicated in Table 5, the (r4, t2) configuration shows a considerable performance drop compared to both the (r3, t3) and (r4, t3) settings.
>
> The angle should have higher precision to ensure that the differences in rotation angles between token positions are as distinguishable as possible, because RoPE introduces positional information based on the differences in these rotation angles.
>
> Specifically, for the lower dimensions of RoPE, after increasing the token position by a group size (g), they undergo multiple full cycles (i.e., the input is rotated by more than 2π). As a result, the quantization of these high-frequency signals in the lower dimensions of RoPE requires higher precision.
>
> Our experimental results in Table 5 also validate the above point.
>
> ---

---

> > ### Comment · Reviewer_FV6N · 2025-08-05
> > **Official Comment**
> >
> > >We find your question interesting, but in this work we primarily lies in activation quantization (e.g., KV cache quantization) rather than weight quantization.
> >
> > Thank you so much for your reply. However, I am not convinced by this answer as the rotation-based methods have been applied to KV-cache quantization also (see A.3 in [1]). In general, rotation-based schemes can improve any kind of quantization (including weights, activations, and caches).
> >
> > >Q3: Ablation on the group size.
> >
> > Thanks for the new results. I agree that those show the accuracy of PolarQuant.
> >
> >
> > >From this analysis, we can see that the overhead introduced by the new key quantization is only 0.6% in comparison to the overall generation cost.
> >
> >
> > Thanks for your results on the runtime. However, the case you evaluated has a small sequence length and maybe KV-caches are not needed (sequence len=256). I would be happy if we could see an analysis on larger sequence lengths (>=2048) and/or some overhead analysis based on the sequence length. In anycase, thanks for providing more results.
> >
> >
> >
> > >The angle should have higher precision to ensure that the differences in rotation angles between token positions are as distinguishable as possible, because RoPE introduces positional information based on the differences in these rotation angles.
> >
> > Thank you so much for providing more information. Yeah, this makes sense to me (and again, suggests using rotation-based methods as well).
> >
> > In conclusion, I found the paper interesting and valuable. However, the replies have not convinced me to change (increase or decrease) my score. I will stick to my previous score.
> >
> > Thanks again for answering my questions.
> >
> > Reference
> > - [1] https://arxiv.org/pdf/2404.00456

---

> > > ### Author Response · Authors · 2025-08-06
> > >
> > > We sincerely appreciate your engagement in our discussion. Your feedback has been of great value in helping us refine our work. We hope our response addresses your concerns.
> > >
> > > ---
> > >
> > > > Compatibility of PolarQuant with rotation-based method
> > >
> > > Thanks for your response. We may have a misunderstanding regarding your original question.
> > >
> > > We'd like to explain the potential difficulties of mixing PolarQuant with rotation-based methods like QuaRot, and hope that can help address your concerns.
> > >
> > > 1. Can we fuse Hadamards of size 2 into the weights of W_k ？
> > >
> > > In the PolarQuant setup, we encode 2D subvectors of the key cache with their radii and angles, which are decoupled and quantized seperately.
> > > Hadamards of size 2 would simply rotate the 2D vectors by a common angle. This equals to adding a bias to all the angle values, and will not improve the quantization effect.
> > >
> > > 2. Can we directly apply PolarQuant to the QuaRot rotated key cache?
> > >
> > > Due to the presence of RoPE, QuaRot rotates the post-RoPE queries and keys using a head-wise Hadamard.
> > > This disrupts the distribution of the key activations in the two dimensions that are rotated together, and there are no well-structured patterns required for polar transformation.
> > >
> > > We appreciate reviewer's insightful suggestion of using rotation-based methods.
> > > We are actively seeking to find a way to combine PolarQuant with it in the future.
> > > Thanks again.
> > >
> > > ---
> > >
> > > > Breakdown time analysis of key cache quantization with longer input length.
> > >
> > > Thank your for the suggestion.
> > > We provide the overhead of getting polar coordinates for inputs with longer sequence length (>=2048). We provide a breakdown time analysis of key cache quantization with a batch size of 1, an input length of 4096, and a generation length of 4096:
> > >
> > > | Operation | | Time (s) |
> > > |-|-|-|
> > > | Decode | All   |  205.24s |
> > > | Quant  | Others|   0.46s  |
> > > |				 | Tan2  |   0.10s  |
> > >
> > > We find that the quantization time overhead is almost identical to the previous experimental results.
> > > This is because the primary quantization cost is incurred during the decoding phase, where new keys will be quantized on the fly, while the quantization cost of pre-filling is negligible.
> > >
> > > We believe the length used in this experiment meets your requirements. We hope the explanation provided above resolves your concerns!

---

### Official Review · Reviewer_bUkd · 2025-06-30

**Clarity:** 3
**Significance:** 3
**Originality:** 3
**Rating:** 4
**Confidence:** 4

**Summary:**

This paper proposes to quantize KV cache through effectively addressing outlier challenges via polar transformation.
Specifically, the authors observe that although the individual axis may contain outliers, they collectively form stable circular patterns, making quantization of the original outliers easier.
Experimental results include models like Llama and Qwen, showing competitive performance with 3-4 bit quantization and up to 3.18x speedup.

**Questions:**

1. Why does Table 4 show PolarQuant using less memory than KIVI despite lookup tables?
2. How does group size $g$ (Sec 3.2) affect performance? With small $g$, there might not be outliers in each group. Will PolarQuant still maintain its superiority under that circumstance?

**Ethical Concerns:**

["NO or VERY MINOR ethics concerns only"]

**Final Justification:**

Most of my concerns have been addressed by authors' reply. Given its contributions, I choose to maintain my original score.

**Limitations:**

Yes

**Quality:**

3

**Strengths And Weaknesses:**

Strengths:
1. Polar transformation novelly exploits RoPE-induced circular patterns (Fig 1b) to mitigate channel-wise outliers.
2. Polar transformation enables decoding acceleration through lookup tables, speeding up attention computation.
3. Comprehensive evaluation on long-context (LongBench), normal context (GSM8K) and reasoning (DeepSeek-R1) across multiple model families.
4. Clear methodology and abundant implementation details.

Weaknesses:
1. Figure 1a claims to show "significantly larger activations" but lacks visual evidence of outliers (no obvious outliers in channel dimension).
2. Given that key value quantization aims to tackle memory bottleneck, the lookup table in PolarQuant incurs extra memory overhead, so it is unclear why memory usage of PolarQuant is smaller than KIVI quant (shown in Table 4).

---

> ### Author Rebuttal · Authors · 2025-07-31
>
> We sincerely appreciate your valuable comments! We hope our response will help address your concerns.
>
> ---
>
> **W1**: Figure 1 needs refinement in presentation.
>
> Thank you for the valuable suggestion to improve the presentation of our paper.
> We will choose more contrasting colors to better highlight the presence of outliers, thereby enhancing the overall presentation of the paper.
>
> ---
>
> **W2 & Q1**：Why does Table 4 show PolarQuant using less memory than KIVI despite lookup tables?
>
> > First of all，here is a misunderstaning that, the lookup table in PolarQuant incurs extra memory overhead.
>
> We want to clarify that the lookup table is actually not cached; instead, it is computed on the fly at block level during decoding (For more details on the implementation, please refer to Appendix A).
> As a result, therotically there should be no obvious difference in memory overhead for PolarQuant and KIVI when the same bit budget is used.
>
> > Second, your observation is insightful that the setups in Table 4 show differences in memory usage, and there are some theoretical discrepancies.
>
>  - For the second comparison group (KIVI-2 & Polar-33):
>
> Polar-33, has higher memory usage than KIVI2 (with more than 1GB at 128K), which is due to the fact that we use our customized 4-bit kernel for Polar-33 in our implementation, introducing 1 bit of redundant memory waste.
>
> The 3-bit kernel implementation requires additional integer unpacking and is complicated to implement, so we use our 4-bit kernel as a replacement.
>
>  - For the third comparison group (KIVI-4† & Polar-44†):
>
> We assume that some hardware issues might be the cause and are currently investigating.
>
> In Polar_44†, we use our kernel implementation for key and KIVI kernel for value.
> In KIVI-4†, the dequantization and matmul operations for both the key and value share the same kernel.
> If the intermediate results by repeated kernel calls are not released, GPU memory usage may accumulate continuously, causing peak memory usage to rise.
>
> We are actively working on pinpointing the memory differences mentioned above. Thank you for your question!
>
> ---
>
> **Q2**: Ablation studies on group size g ?
>
> Thanks for your suggestion! We conduct an ablation study exploring different group size using Llama-3.1-8B-instruct.
>
> The table below presents the performance of PolarQuant and KIVI on LongBench with a 4-bit setting (* indicates better preformance with same group size):
>
> | model | group size | bits | LongBench (Avg.) |
> |-|-|-|-|
> | KIVI  | 32         | 5.00 |     49.48        |
> | KIVI  | 64         | 4.50 |     49.47*       |
> | KIVI  | 128        | 4.25 |     49.36        |
> | KIVI  | 256        | 4.13 |     49.52        |
> | Polar | 32         | 5.00 |     49.50*       |
> | Polar | 64         | 4.50 |     49.33        |
> | Polar | 128        | 4.25 |     49.39*       |
> | Polar | 256        | 4.13 |     49.58*       |
>
> This suggests that PolarQuant performs competitively with KIVI across different group sizes.
>
> > With small g, there might not be outliers in each group. Will PolarQuant still maintain its superiority under that circumstance?
>
> We would like to make two points in response:
>
> (1) The quantization parameters are of size 32/g bits. It is clear that smaller group sizes introduce an additional cost.
>
> (2) Considering the overhead of quantization parameters caused by a smaller g and the sufficiently good performance with a relatively large g, we recommend using 128, which is the setting used in our paper.
>
> ---

---

> > ### Comment · Reviewer_bUkd · 2025-08-05
> >
> > Thank you for your responses.
> > The concerns and questions have been addressed by the explanations and the additional experimental results.
> > I will remain my scores.

---

### Official Review · Reviewer_Mgry · 2025-07-03

**Clarity:** 4
**Significance:** 3
**Originality:** 4
**Rating:** 5
**Confidence:** 4

**Summary:**

This paper introduces PolarQuant, a novel quantization method for the K-cache in large language models. The method leverages a polar coordinate transformation to address the challenge of outliers in key vectors, which traditionally hinder effective quantization. By encoding key vectors as quantized radii and angles, PolarQuant achieves both memory efficiency and decoding acceleration, while maintaining high performance across various tasks and model architectures.

**Questions:**

- Recent models like Qwen3 have adopted a pre-RoPE normalization layer (https://github.com/huggingface/transformers/blob/main/src/transformers/models/qwen3/modeling_qwen3.py#L185), which retains the polar characteristics of RoPE while reducing the quantization burden. Would PolarQuant still be applicable or effective for such models?

- How was the value cache (V-cache) quantization method chosen? Was it selected because KIVI demonstrated the best performance among existing approaches

- Given that PolarQuant relies on the structured circular patterns induced by RoPE, how sensitive is the method to variations in RoPE configurations (e.g., different base frequencies or dimensional pairings)? Could this affect its generalizability across diverse model architectures or custom RoPE implementations?

**Ethical Concerns:**

["NO or VERY MINOR ethics concerns only"]

**Limitations:**

-

**Paper Formatting Concerns:**

-

**Quality:**

4

**Strengths And Weaknesses:**

Strength:

- The use of polar coordinates to mitigate outlier effects in key vectors is a novel and elegant idea. The authors show that outliers often appear in only one of the two dimensions rotated by RoPE, and that these pairs form smooth circular patterns in polar space, making them more amenable to quantization.
- The authors provide detailed implementation and reproducibility information, which is commendable.

---

> ### Author Rebuttal · Authors · 2025-07-31
>
> We sincerely appreciate your valuable comments and hope our responses address your concerns.
>
> ---
>
> **Q1**: More diverse model architectures like Qwen3 ?
>
> In our experiments, we focus on popular model architectures, such as Llama and Qwen, as we believe they have demonstrated the broad applicability of PolarQuant across different architectures.
>
> Your observation is very insightful, and we agree that the pre-RoPE normalization holds significant value.
> Given the time constraints, we have decided to leave the exploration of integrating PolarQuant with Qwen3 for future work.
> Thank you for your valuable suggestion.
>
> ---
>
> **Q2**: How was the value cache quantization method chosen ?
>
> Our paper focuses on addressing the challenges of quantizing keys, and is orthogonal to specific value quantization algorithms, as demonstrated in the following aspects:
>
> 1. In Tables 1, 2, 3, 5, we maintain the value cache in full precision. This effectively isolate the influence of value quantization algorithms and clearly demonstrate the effectiveness of our method in addressing the challenges of key quantization.
> 2. After establishing the effectiveness of our method, in Table 6, we employ KIVI to quantize values. This step is taken to showcase the compatibility of PolarQuant with value quantization algorithm, thus making our paper more comprehensive. **We selected KIVI simply because it is a classic method and a frequently compared baseline in the field.**
>
> Taken together, these twofold experiments demonstrate that PolarQuant works well with both full-precision values to low-bit quantized  values using classic methods—without performance degradation.
>
> ---
>
> **Q3**: How sensitive is PolarQuant to the RoPE configuration? Could this affect its generalizability across diverse model architectures or custom RoPE implementations?
>
> Thank you for the insightful question. We have experimented with diverse model architectures, each with a unique RoPE setup, including Llama-2-7B-Chat (base frequency 10000), Llama-3.1-8B-Instruct (with rope base as 500000), and Qwen-2.5 (with rope base as 1000000).
>
> In addition, we show that PolarQuant is adaptable to different RoPE variants, which addresses your concern about its generalization to custom RoPE implementations. Specifically, we perform experiments with NTK-aware Scaled RoPE on Llama-2-7B-Chat to demonstrate this.
>
> |Method|LongBench (avg.)|
> |-|-|
> |NTK| 32.15 |
> |NTK-Polar44| 32.44|
>
> The experimental results indicate that PolarQuant can adapt well to the customized RoPE configuration, hope this can address your concern.

---

### Official Review · Reviewer_Jun3 · 2025-07-03

**Clarity:** 3
**Significance:** 2
**Originality:** 3
**Rating:** 4
**Confidence:** 4

**Summary:**

PolarQuant aims to accurately quantize the key cache to low precision by leveraging a polar representation for the keys. Their approach is designed to address the impact of the RoPE embedding (which rotates neighboring channels in key activations). They observe that when representing pairs of channels in polar form, the outlier channels in keys (which are typically coupled with a non-outlier channel) become easier to quantize since they have a consistent norm along the sequence length dimension. By quantizing keys in polar form, they achieve superior efficiency (in terms of improved accuracy for the same bit width) relative to existing methods. They also provide an efficient system implementation that exploits table lookups to accelerate inference, showing the efficiency gains of their approach through both kernel-level and end-to-end benchmarking.

**Questions:**

- What is the overhead of quantizing new keys online during inference using this method - is it negligible?
- Do you have any analysis demonstrating the relative higher sensitivity of the key cache to low-precision quantization (compared with values)? Also, is the key cache still more sensitive than the value cache even after applying the polar quant method?

**Ethical Concerns:**

["NO or VERY MINOR ethics concerns only"]

**Final Justification:**

The rebuttal clarified my concerns around the key cache vs value cache sensitivity and overhead of quantizing new keys.

With respect to the concerns around efficiency of fused-RoPE attention, I am not convinced (based on the speedups of more optimized fused-RoPE kernels) that there is a reason that this can't be efficiently implemented with low-bit quantization. However, I understand the author's feedback that there aren't existing kernel implementations for this and that it is difficult to prove definitively that this can't be implemented efficiently, and that they compared with the best available open-source method. As such, I will raise my score to a 4.

**Limitations:**

Yes

**Quality:**

3

**Strengths And Weaknesses:**

Strengths:
- They propose a novel polar transformation to tackle the effects of RoPE on the key distributions (which counteracts the effect of the rotation on widening the range of outlier channels).
- They present convincing evaluation across multiple models (llama3.1 and qwen2.5) and datasets (LongBench + GSM8K) as well as comparisons with existing strong baselines like KIVI/QJL.
- They present a lookup table mapping for efficient decoding as well as custom Triton kernels which show substantial latency gains with their method. Their speedup evaluation includes kernel-level and end-to-end evaluation.

Weaknesses:
- The insight that the key cache has outlier channels with consistent magnitude pre-rope and that the magnitude gets rotated with the neighboring channel (leading to varying magnitude along the sequence length) is not novel relative to prior work. Although they argue that their method is superior to applying RoPE during inference as the key cache is loaded (which is an alternative strategy to account for the rotation), they do not justify that the overhead of performing recomputation using RoPE is substantial (particularly since KV cache loading is memory bandwidth bound, this overhead may potentially be hidden / overlapped with memory loads).
- The fast decoding method using lookup tables makes sense, but is not entirely novel in the context of KV cache quantization - for example, previous non-uniform quantization methods like KVQuant have leveraged lookup tables for fast dequantization.
- Their method focuses mainly on efficient key quantization; however, even if we can push keys to low bit precision, we will still be limited by the precision with which we can represent values. A more detailed analysis of challenges with value quantization is not presented. Alternatively, some ablations demonstrating that the relative difficulty of value quantization relative to key quantization is low would be useful.

---

> ### Author Rebuttal · Authors · 2025-07-31
>
> Thank you for valuable feedbacks! Hope our explanation helps address your questions.
>
> ---
>
> **W1:** The authors do not justify that the overhead of RoPE recomputation in previous work is substantial, particularly since KV cache loading is memory bandwidth bound.
>
> Thank you for the suggestion.
>
> > First of all, we contend that **the overhead of RoPE recomputation is substantial**.
>
> 1. Perspective of Theoretical Analysis
>
> KVCache loading in full precision is memory-bound [1, 2], but low-bit KV cache loading with matrix multiplication dequantization does **NOT** exhibt the same on more advanced GPU architectures.
>
> KVQuant utilizes A6000 to analyze kernel efficiency, whereas our study uses the A800, which is a more advanced GPU and reflects a more practical scenario.
> We monitor the Memory Throughput and Compute Throughput of the vecquant4matmul kernel of KVQuant on A800.
> The NCU report indicates that such kernel implementations exhibit a computation-bound.
>
> 2. Perspective of Experimental Results
>
> We conduct further experiments to empirically confirm that the overhead of RoPE recomputation is substantial.
>
> On A800, we benchmark the latency of the KVQuant kernel implementation, as well as its end-to-end throughput performance in Transformer.
> For batch size, we identify the maximum supported batch size for a generation length of 4K, with other experimental settings following those outlined in Section 4.2.
>
>
> The experimental results are as follows:
>
> Latency comparisons  of kernels across  context lengths
>
> | Method \ Length |  4k  |   8k  |  16k |  32k  |
> |-|-|-|-|-|
> |	Fp16             | 0.35 |  0.66 | 1.30 |  2.56 |
> | KVQuant          | 1.23 |  2.22 | 4.29 |  8.44 |
> | KIVI4		         | 0.69 |  1.38 | 2.75 |  5.47 |
> | PolarQuant44     | 0.33 |  0.61 | 1.16 |  2.26 |
>
> Throughput comparison
> |   Method             |  Fp16   |  KVQuant | KIVI4  | PolarQuant44 |
> |-|-|-|-|-|
> | Throughput (token/s) |  129.78 |  135.97  | 140.94 |    157.16    |
>
>
> These two aspects will demonstrate that the overhead of RoPE recomputation is substantial.
>
> > We hope the reviewer can reconsider the novelty of PolarQuant, given that the overhead of RoPE recomputation is substantial.
>
> 1. We highlight the limitations of KVQuant, where in real-world scenarios like on A800, **the recomputation of RoPE is substantial**, leading to efficiency issues.
>
> 2. PolarQuant introduces a **novel polar transformation approach to key cache quantization** that avoids this recomputation.
>
> Similar to KVQuant, our method utilizes pre-RoPE key properties. However we argue that the polar transformation is an innovative approach to preserve pre-RoPE benefits without the recomputation burden, leading to faster inference across a wider range of applications.
> We bilieve this is one of our contributions.
>
> Thank you for the insightful suggestion. We hope you will reconsider your evaluation of the novelty of our work.
> We will include the necessary discussion in the revision to clarify any potential misunderstandings.
>
> **References:**
>
> [1] SqueezeLLM: Dense-and-Sparse Quantization
>
> [2] KVQuant: Towards 10 Million Context Length LLM Inference with KV Cache Quantization
>
> ---
>
> **W2**: The novelty of PolarQuant's use of lookup table.
>
> We highlight our novelty and contribution by distinguishing (1) the usage of lookup tables and (2) the effectiveness of lookup tables ultization between PolarQuant and KVQuant.
>
> > Differences in lookup table usage scenarios
>
> - KVQuant utilizes a **lookup table to dequantize the key cache** and restores the positional information of key values via RoPE recomputation.
> - PolarQuant avoids RoPE recomputation; instead, it directly constructs **a lookup table for QK product** on the fly to achieve acceleration.
>
> > Differences in effectiveness
>
> As highlighted in our response to W1, the cost of RoPE recomputation has become non-negligible on advanced GPUs, emerging as a bottleneck that limits KVQuant’s efficiency.
> On the other hand, PolarQuant offers improved acceleration.
>
>
>
> In summary, our use of the lookup table is neither a follow-up nor an incremental improvement over KVQuant. While lookup tables are a common technique in quantization [1, 2, 3], many aspects of PolarQuant are novel:
>
> - The challenging problem we address.
> - The rationale behind our application of lookup tables.
> - The specific objects our tables target.
>
> **References:**
>
> [1] KVQuant: Towards 10 Million Context Length LLM Inference with KV Cache Quantization
>
> [2] PQCache: Product Quantization-based KVCache for Long Context LLM Inference
>
> [3] PCDVQ: Enhancing Vector Quantization for Large Language Models via Polar Coordinate Decoupling
>
> ---
>
> **W3 & Q2**: Is Key cache more sensitive to low-precision quantization than Value cache?
>
> Yes. In Section 5.2, we combine 4-bit key quantization (PolarQuant_44) with 2-bit value quantization and observe minimal performance degradation.
>
> Additionally, we test value quantization (KIVI with group size 128) on LongBench while keeping the key cache at full precision and resulting in no performance drop. However, when quantize key in same bit and keep value in full precision, the performance greatly drops.
> | Method        | Key bits | Value bits | LongBench (Avg.) |
> |-|-|-|-|
> | BF16          | 16       | 16         | 49.26            |
> | KIVI          | 16       | 4.25       | 49.54            |
> | KIVI          | 16       | 2.25       | 49.30            |
> | KIVI          | 2.25     | 16         | 47.73            |
>
> These results suggest that:
>
> 1. Existing quantization methods are already sufficiently effective in value quantization.
> 2. Key quantization is more sensitive to precision changes than value quantization in current methods.
> As a result, key quantization presents more challenges than value quantization.
>
> Thank you for the suggestion, we will include this discussion in our revised version.
>
> ---
>
> **Q1:** Overhead of quantizing new keys ?
>
> The following is the detailed time breakdown for the PolarQuant_44 during the decoding phase (with a batch size of 1, input length of 256, and generation length of 4096):
> | Operation | Time (s) |
> |-|-|
> | Decode    |   91.07    |
> | Quant     |   0.55    |
>
> From this, we can conclude that the overhead of quantizing new keys is 0.6% compared to other operations.
> Quantization is performed every n steps, causing only a slight overhead during decoding, which explains why it is negligible.
>
> ---

---

> > ### Comment · Reviewer_Jun3 · 2025-08-03
> >
> > I appreciate the detailed rebuttal from the authors.
> >
> > While the results clearly show that the PolarQuant method provides greater speedups than the KVQuant implementation, I am not convinced from the provided data that the speed differences observed between these methods are due to fundamental limitations of pre-RoPE key quantization (and not due to other inefficiencies in the KVQuant implementation / algorithm). For state-of-the-art attention kernels like FlashInfer, "Experiments on various platform and settings show that FlashInfer’s Fused-RoPE Attention kernel can apply RoPE on the fly with negligible overhead." (https://flashinfer.ai/2024/02/02/introduce-flashinfer.html). It also doesn't really make sense to me that the RoPE operation would be compute bound - the ratio of peak compute to memory bandwidth on GPUs is typically in the 100s, and the number of arithmetic operations required for performing RoPE is ~5 per element in the vector. I am wondering if this is an artifact of the particular kernel implementation, and not a fundamental issue with applying RoPE on-the-fly. I would appreciate it if the authors could provide some explanation for why they do not believe that applying RoPE on-the-fly is possible at all.
> >
> > The final two responses (key cache vs value cache sensitivity and overhead of quantizing new keys) addressed my concerns.  With respect to the novelty of the lookup table method, the response has clarified the difference between the use of lookup tables in this work versus prior methods.

---

> > > ### Author Response · Authors · 2025-08-05
> > >
> > > Thank you for your quick response. We hope the following reply will help clarify some of your concerns:
> > >
> > > ---
> > >
> > > > Can the speed differences caused by other inefficiencies in KVQuant implementation?
> > >
> > > We appreciate the reviewer recognizing that the PolarQuant method provides greater speedups than the KVQuant implementation.
> > >
> > > We want to clarify that we have made efforts to minimize the potential impact of inefficiencies in the KVQuant implementation on the final speed.
> > > During the rebuttal phase, we use the official KVQuant kernel [1] to exclude the interference of other algorithm designs in KVQuant (e.g., per-vec dense-and-sparse quantization) on speed, apart from the on-the-fly RoPE recomputation.
> > > This kernel implementation [1] only includes key dequantization with lookup tables and the fused-RoPE attention.
> > > Therefore, we believe that the on-the-fly RoPE recomputation in KVQuant is the factor affecting the efficiency.
> > >
> > > [1]  vecquant4matmul_nuq_perchannel_transposed_rope_mha_batched_fused_opt
> > >
> > > ---
> > >
> > > > Is applying RoPE on the fly possible in low-bit quantization attention ？
> > >
> > > We want to clarify that "the authors don't believe applying RoPE on-the-fly is possible".
> > >
> > > As mentioned in our rebuttal, we agree that the overhead of applying key cache RoPE on-the-fly is negligible in full precision, e.g in FlashInfer.
> > > However, the overhead of RoPE-Fusion in low-bit quantization attention remains underexplored.
> > >
> > > Few works perform on-the-fly RoPE recomputation in low-bit key quantization, and KVQuant is the most widely discussed.
> > > Therefore, we analyze it based on KVQuant's kernel implementation.
> > > We believe that each paper’s authors are responsible for their methods and provide the best code implementation.
> > > Our experiments, based on the KVQuant kernel, suggest that existing efforts to apply RoPE on-the-fly present challenges in low-bit quantization attention.
> > > We also look forward to the community’s development of more efficient implementations.

---

### Comment · Area_Chair_rZ6e · 2025-08-05
**[From AC] Reviewer Discussion Reminder**

Dear Reviewers,

Thank you for your time and effort in reviewing the paper.

As the reviewer-author discussion period ends on August 6 at 11:59 PM AoE, please take a moment to acknowledge the rebuttal and engage in the discussion if you haven’t.

Thank you again for your contributions to the review process.

Best,\
Area Chair

---

### Note · Authors · 2025-08-15

Dear Reviewers, AC, SAC, and PC,

We sincerely thank all chairs and reviewers for your feedback that has helped improve our paper.
Below we summarize the strengths recognized by reviewers and our responses to their concerns:

## Strengths Recognized by Reviewers

In this work, we propose PolarQuant, a novel approach that addresses the challenge of outliers in key quantization.
**All reviewers recognize our use of polar transformation to handle these outliers as both innovative and insightful**.

Building on that, a novel decoding acceleration method is developed.
**Reviewers consistently find that PolarQuant provides greater speedups than baseline methods, demonstrating its efficiency**.

Furthermore, we validate PolarQuant through extensive experiments across diverse tasks and models.
The reviewers note that our method **offers a detailed implementation and a thorough evaluation**, which fully demonstrates the completeness and robustness of PolarQuant (Reviewers Mgry, bUkd, FV6N).

We believe our contribution is valuable to the community, given the heavy burden of KV-cache management in LLMs.

## Addressing Reviewers' Concerns

We note that Reviewers Jun3 and FV6N raised new questions late in the discussion period and did not respond to our subsequent clarifications.

Below is a summary of our key responses to these two reviewers.

> Regarding Reviewer Jun3

We appreciate the reviewer's recognition that PolarQuant provides greater speedups than KVQuant.

We want to clarify that KVQuant is the most widely discussed work that performs RoPE recomputation in low-bit key quantization, and we have made efforts to mitigate the inefficiencies in the KVQuant implementation.
Based on our experiments with KVQuant, we observe that the overhead from RoPE recomputation is substantial, and PolarQuant helps avoid this overhead.

We have addressed Reviewer Jun3's other concerns and hope this final response resolves the remaining question.


> Regarding Reviewer FV6N

(1) Our additonal quantitative analyses show that the overhead of obtaining polar coordinates is negligible on larger sequence lengths. Reviewer Jun3 also had a similar concern, which has been resolved.

(2) We explain the difficulties of mixing PolarQuant with rotation-based methods like QuaRot and leave it to future work to explore this possibility.


---

We sincerely thank all chairs and reviewers for your valuable input and hope that we have addressed your concerns.

Sincerely,

Authors

---

### Decision · Program_Chairs · 2025-09-17

**Decision:**

Accept (poster)

**Comment:**

This paper introduces PolarQuant, a method for key-cache quantization using polar coordinates to address RoPE-induced outliers, combined with a lookup-table–based decoding acceleration.

The approach is well-motivated and empirically validated across multiple models and datasets, demonstrating notable speedups and robustness. Reviewers highlight the innovative use of polar transformation and its practical acceleration benefits.
Some clarifications remain regarding pre-RoPE normalization, sensitivity to RoPE configurations, recomputation with lookup tables, and ablations on group size and base frequencies, which should be included in the final version.

Overall,  the work presents a creative, technically sound, and practically impactful contribution.

Decision: Accept